# DEBIASED IMBALANCED PSEUDO-LABELING FOR GENERALIZED CATEGORY DISCOVERY

## ABSTRACT

Generalized Category Discovery (GCD) is a challenging task that aims to recognize seen and novel categories within unlabeled data by leveraging labeled data. Designing a prototype classifier to identify unlabeled samples instead of relying on traditional time-consuming clustering is well recognized as a milestone in GCD. However, we discover there exists a bias in this classifier: some seen categories are mistakenly classified as novel ones, leading to imbalanced pseudo-labeling during classifier learning. Based on this finding, we identify the low discriminability between seen and novel prototypes as the key issue. To address this issue, we propose DebiasGCD, an effective debiasing method that integrates *dynamic prototype debiasing* (DPD) and *local representation alignment* (LRA). DPD dynamically maintains inter-prototype margins, encouraging the network to strengthen the learning of class-specific features and enhance prototype discrimination. Additionally, LRA promotes local representation learning, enabling DPD to capture subtle details that further refine the understanding of class-specific features. In this way, it successfully improves prototype discriminability and generates more reliable predictions for seen classes. Extensive experiments validate that our method effectively mitigates pseudo-labeling bias across all datasets, especially on fine-grained ones. For instance, it delivers a 10.7% boost on 'Old' classes in CUB. Our code is available at https://anonymous.4open.science/r/DebiasGCD-34F0.

## 1 INTRODUCTION

Deep learning has achieved great success in computer vision (He et al., 2017; Ren et al., 2015; Ronneberger et al., 2015; Szegedy et al., 2015; Redmon et al., 2016; Dosovitskiy et al., 2021; Caron et al., 2021). However, these achievements are based on a close-world assumption, where test data shares the same classes as the training data. This assumption fails in open-world scenarios (Geng et al., 2021; Han et al., 2022; Zhang et al., 2023a; Wang et al., 2023b), where unlabeled data may encounter unseen categories. To address this, a new paradigm called *Generalized Category Discovery* (GCD) Vaze et al. (2022b) has been proposed and is gaining increasing attention. GCD requires models to recognize seen and novel categories in unlabeled data by leveraging knowledge from labeled data, making it suitable for open-world scenarios with vast amounts of unlabeled data.

There are mainly two existing approches for GCD: $k$-means clustering (Vaze et al., 2022b; Fei et al., 2022; Zhao et al., 2023; Pu et al., 2023) and prototype classification (Wen et al., 2023; Wang et al., 2024). The former identifies unlabeled samples by clustering their representations. However, it often becomes computationally expensive with larger datasets due to the quadratic complexity of clustering algorithms. Instead, Wen et al. Wen et al. (2023) adopts the latter approach, and proposes SimGCD, replacing the clustering-based approach with a classifier. Specifically, they found that using a classifier directly in GCD led to overfitting on seen categories, causing novel categories to be misclassified as seen ones. To fix this, they introduced a mean-entropy-maximization regularizer (Assran et al., 2022) to activate novel prototypes learning and improve pseudo-label reliability for the prototype classifier. As a result, SimGCD replaces the time-consuming clustering with a prototype classifier and achieves competitive performance, establishing itself as a robust baseline in GCD.

Although SimGCD successfully mitigates the overfitting bias on seen categories, a detailed investigation reveals that there also exists a new bias in the classifier: some seen categories are mistakenly classified as novel ones, leading to imbalanced pseudo-labels during classifier learning. This issue

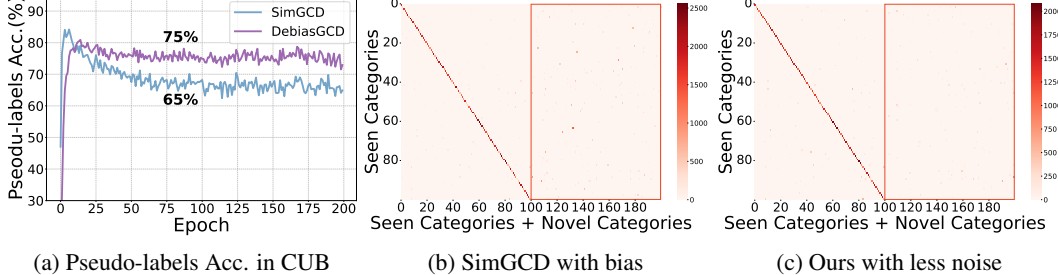

Figure 1: Pseudo-labeling of seen samples in CUB (Wah et al., 2011): (a) shows that DebiasGCD outperforms SimGCD by 10% in pseudo-labeling accuracy. (b) and (c) display the distribution of specific pseudo-label assignments, where darker points indicate more assignments. In (b), the darker points in the red rectangle highlight the bias, where seen category samples (y-axis, 0-100) are incorrectly labeled as novel categories (x-axis, 100-200). In (c), DebiasGCD shows lighter colors, indicating reduced imbalanced pseudo-labeling.

is clear when tracking the pseudo-labeling of seen categories. As shown in Fig. 1a, the blue curve indicates that the pseudo-label accuracy in SimGCD fluctuates around 65% to the end. Fig. 1b further details the pseudo-label assignment, where the diagonal line indicates correct classification, and the points with deep color in the red rectangle highlight seen samples (Class id $< 100$) misclassified as novel (Class id $\geqslant 100$). Consequently, SimGCD struggles with accurate pseudo-labeling of seen categories during training. We attribute this bias to two reasons: *indiscriminate prototype learning* and *simple representation alignment*. First, models trained with cross-entropy on labeled data with only seen categories lack the guidance necessary to distinguish seen and novel classes. Therefore, the prototypes of seen categories lack the discriminability to recognize samples, resulting in biased pseudo-labeling. Second, the previous prototype classifiers (Wen et al., 2023; Vaze et al., 2023) only utilize global representations (i.e. class token) for prototype learning with self-distillation (Caron et al., 2021; Assran et al., 2022), ignoring the local ones (i.e. patch tokens) beneficial for classification. This omission leads to insufficient feature guidance, hindering prototypes from learning discriminative representations and compromising pseudo-label quality.

To tackle the above issues, we propose an effective method called **DebiasGCD** to calibrate bias between prototypes. First, we introduce a *dynamic prototype debiasing* (**DPD**) technique to reduce the prediction bias induced by prototypes dynamically. This technique maintains inter-prototype margins between classes, enhancing the network's ability to learn more class-specific features, and reinforcing the prototype discrimination. As a result, the classifier can generate more reliable pseudo-labels. Second, we propose a *local representation alignment* (**LRA**) module to align the local representations of different sample views in semi-supervised learning. This helps the classifier learn detailed class-specific representations and facilitates the optimization of DPD. Additionally, we also adopt *strong* and *weak* augmentation for instances following Vaze et al. (2023); Sohn et al. (2020). Finally, by combining DPD and LRA, DebiasGCD creates clear boundaries for class prototypes, especially for seen and novel classes, thus mitigating imbalanced pseudo-labeling.

To evaluate the effectiveness of our method, we conduct extensive experiments on six datasets, including fine-grained and generic object classification datasets. Our approach significantly reduces pseudo-labels bias, outperforming SimGCD (Wen et al., 2023) by **10.7%/5.2%** on the 'Old' and 'New' classes in CUB, respectively. Meanwhile, DebiasGCD prevents a 10% drop in pseudo-label accuracy, as shown in Fig. 1a. Fig. 1c further shows DebiasGCD produces less noise in the red rectangle compared to SimGCD (Fig. 1b), indicating successful mitigation of imbalanced pseudo-labels.

In summary, our key contributions are as follows: **1)** We investigate and discover a bias exists in the classifier that leads to imbalanced pseudo-labeling in GCD semi-supervised learning; **2)** We propose effective debiased imbalanced pseudo-labeling, DebiasGCD, which combines DPD and LRA to expand the distances between prototypes and extracts more local discriminative representation features for recognizing instances in unlabeled data; **3)** Extensive experimental results demonstrate our proposed debiased learning removes bias effectively, improving the performance by a large margin e.g., **10.7%/5.2%** on 'Old' and 'New' classes in CUB, respectively.

## 2 RELATED WORK

**Semi-supervised Learning** (SSL) tackles the issue of limited labeled data by integrating unlabeled data from pre-defined classes (Goodfellow et al., 2014; Laine & Aila, 2017; Huang et al., 2021). Most SSL methods adopt techniques like consistency-based regularization (Menon et al., 2021; Tarvainen & Valpola, 2017; Miyato et al., 2019; Xie et al., 2020), pseudo-labeling (Sohn et al., 2020; Zhang et al., 2022; Yu et al., 2023), and transfer learning (Chen et al., 2020b). Notably, pseudo-labeling is an effective baseline using a weak augmented view's prediction as a pseudo-label for a strong view Berthelot et al. (2020); Sohn et al. (2020). Meanwhile, Wang et al. (2023a); Xu et al. (2021); Zhang et al. (2021) improves performance by adjusting thresholds to select high-quality pseudo-labels. However, traditional SSL methods assume the labeled and unlabeled data share the same distribution, limiting its effectiveness in open-world settings. Recently, SSL has been extended to out-of-distribution (OOD) detection (Huang et al., 2023; Yu et al., 2023; Du et al., 2023; Zheng et al., 2023). For instance, FlatMatch Huang et al. (2023) minimizes cross-sharpness for consistent learning, InPL Yu et al. (2023) uses energy-based pseudo-labeling to select pseudo-labeling for OOD data, and ATOL Zheng et al. (2023) employs generated OOD data to improve detection. Increasingly, SSL methods are being adapted for real-world scenarios.

**Generalized Category Discovery** (GCD) Vaze et al. (2022b) aims to train a model that can recognize both seen and novel categories within unlabeled data. Unlike Novel Class Discovery (NCD) (Han et al., 2019; Li et al., 2023; Yang et al., 2023), which treats unlabeled data as entirely new classes, GCD assumes a mix of seen and novel classes. Current GCD methods can be categorized into two paradigms: $k$-means clustering (Vaze et al., 2022b) and learnable classifiers (Wen et al., 2023). First, most works adopt a clustering strategy to learn the representation center to recognize unlabeled instances. XCon Fei et al. (2022) partitions datasets into visually similar sub-datasets using $k$-means clustering, forcing the model to learn fine-grained features. GPC Zhao et al. (2023) uses a Gaussian Mixture Model (GMM) and representation learning to cluster categories. AGCD Ma et al. (2024) incorporates active learning to increase labeled data for clustering. PromptCAL Zhang et al. (2023b) enhances semantic representations with prompt learning and contrastive affinity, but still relies on SemiKMeans clustering (Vaze et al., 2022b). However, these methods are computationally expensive, particularly with large datasets. To address this, parametric classifier methods have emerged. SimGCD Wen et al. (2023) introduces a prototype classifier establishing a new GCD baseline. $\mu$GCD Vaze et al. (2023) utilizes mean-teacher to enhance pseudo-label quality for SSL, building upon SimGCD. It initially trains with the clustering-based GCD (Vaze et al., 2022b) before fine-tuning with the classification head, which is time-intensive. In this paper, we also improve the quality of pseudo-label for SSL in a time-saving manner based on SimGCD.

**Prototypes Learning** regards the class-specific representations in feature space as prototype centers for each category (Snell et al., 2017). In most GCD works, instances are matched to the category with a larger similarity to its prototype. For instance, DPN An et al. (2023) uses a decoupled prototypical network to separate seen and novel categories, aligning them in labeled and unlabeled data to transfer knowledge and capture semantics. TAN An et al. (2024) trains a model to align instances with prototypes and estimate novel prototypes in unlabeled data based on category similarities. Some works combine contrastive learning (Caron et al., 2021) with classifiers to learn the prototypes. SimGCD Wen et al. (2023) first proposes to construct a parametric classifier with prototypes for categories. It employs contrastive learning technique in labeled and unlabeled data to train the model to learn the class discriminative features and then employs self-distillation (Caron et al., 2021; Assran et al., 2022) to generate pseudo-labels for further optimization. Obviously, the parametric classifier relieves the cost of transferring and aligning prototypes and is more effective in recognizing the instances. Unfortunately, the classifier in SimGCD ignores training a set of more discriminative class prototypes, leading to unreliable pseudo-labels.

## 3 PROBLEM STATEMENT AND PRELIMINARIES

**Generalized category discovery (GCD).** Suppose $\mathcal{X}$ is the input space, we assume the labeled dataset $\mathcal{D}^l = \{(\boldsymbol{x}_i, y_i)\} \in \mathcal{X} \times \mathcal{Y}_l$, containing only known categories, and the unlabeled dataset $\mathcal{D}^u = \{(\boldsymbol{x}_i, y_i)\} \in \mathcal{X} \times \mathcal{Y}_u$, which includes both seen and novel categories, where $\mathcal{Y}_l \subset \mathcal{Y}_u$. The objective of GCD is to categorize the samples in unlabeled data $\mathcal{D}^u$, using the labels from known categories ($\mathcal{Y}_l$) and unlabeled data ($\mathcal{D}^u$). Notably, the total number of known and novel categories is

denoted as $K = |\mathcal{Y}_l \cup \mathcal{Y}_u|$, and $K$ is known from prior works (Wen et al., 2023; Fini et al., 2021; Han et al., 2022; Zhao & Han, 2021; Zhong et al., 2021).

**Architecture.** We investigate the problem in the domain of computer vision. Following SimGCD (Wen et al., 2023), we construct a model $f_\theta$ with parameter $\theta$ to map an input image to a label in $\mathcal{Y}_u$. The structure of the model $f_\theta$ includes an image encoder $\Phi$, a projection head $h$, and a classification head $g$. Given a sample $\boldsymbol{x}_i$, its feature representation is $\boldsymbol{z}_i = \Phi(\boldsymbol{x}_i) \in \mathbb{R}^{(n+1) \times d}$ where $n$ is the number of patches, 1 corresponds to the class token, and $d$ is the number of dimensions of the feature space. The number of patches $n = \frac{H \times W}{h \times w}$ where $H$ and $W$ are the height and width of the image, $h$ and $w$ are the patch sizes.

**Representation Learning.** Just as GCD (Vaze et al., 2022b) and SimGCD (Wen et al., 2023), we adopt contrastive learning (Caron et al., 2021) to extract representation features for categorizing unlabeled data. Specifically, we perform supervised contrastive learning (Khosla et al., 2020) on labeled data $\mathcal{D}^l$ and self-supervised contrastive learning (Chen et al., 2020a) on the whole data set. The overall representation loss is denoted as $\mathcal{L}_{rep}(\theta; \mathcal{D})$ (See Appendix).

**Prototype Classifier.** Unlike GCD (Vaze et al., 2022b), which uses a time-intensive clustering-based approach such as $k$-means, SimGCD Wen et al. (2023) designs an effective prototypical classifier $g$ with parameter $\mathbf{W} \in \mathbb{R}^{d \times k}$ based on self-distillation (Caron et al., 2021; Assran et al., 2022). The column vectors in $\mathbf{W} = [w_0, \ldots, w_{k-1}]$ can be regarded as $k$ prototypes, one corresponding to a category (label). Given an input $\boldsymbol{x}_i$, its class token is the first row vector in its feature representation $\boldsymbol{z}_i$, denoted as $\boldsymbol{z}_i^{cls}$. The cosine similarities between the class token and the prototypes are

$$\boldsymbol{s}_i = [s_{0,i}, s_{1,i}, \ldots, s_{k-1,i}] = \left[\langle \mathbf{w}_0, \boldsymbol{z}_i^{cls} \rangle, \langle \mathbf{w}_1, \boldsymbol{z}_i^{cls} \rangle, \ldots, \langle \mathbf{w}_{k-1}, \boldsymbol{z}_i^{cls} \rangle\right] \quad (1)$$

After obtaining the similarities $\boldsymbol{s}_i$, we can perform softmax and imitate the traditional classifier to get the pseudo-logits by:

$$\boldsymbol{p}(\boldsymbol{x}_i) = \frac{\exp\left(\boldsymbol{s}_i/\tau\right)}{\sum_{j=0}^{k-1} \exp\left(\boldsymbol{s}_{i,j}/\tau\right)}. \quad (2)$$

where the logits of views $\boldsymbol{x}$ and $\boldsymbol{x}'$, forwarded to the student and teacher network, correspond to $\boldsymbol{p}_T(\boldsymbol{x}_i)$ and $\boldsymbol{p}_S(\boldsymbol{x}'_i)$, respectively. Then, it employs a standard cross-entropy loss to supervise the learning of the prototype classifier in *all* data in **un**supervised way:

$$\mathcal{L}_{cls}^u(\theta; \mathcal{D}) = -\frac{1}{|\mathcal{D}|} \sum_{\boldsymbol{x}_i \in \mathcal{D}} \langle \boldsymbol{p}_T(\boldsymbol{x}_i), \log\left(\boldsymbol{p}_S(\boldsymbol{x}'_i)\right)\rangle + \langle \boldsymbol{p}_T(\boldsymbol{x}'_i), \log\left(\boldsymbol{p}_S(\boldsymbol{x}_i)\right)\rangle \quad (3)$$

Note that the model also jointly trained utilizing a standard cross-entropy loss on labeled data in supervised way, represented as $\mathcal{L}_{cls}^s(\theta; \mathcal{D}^l)$ along with an entropy regularization, $\mathcal{L}_r(\theta; \mathcal{D})$. Therefore, the total classification loss is denoted as: $\mathcal{L}_{cls}(\theta; \mathcal{D}) = \mathcal{L}_{cls}^s(\theta; \mathcal{D}^l) + \mathcal{L}_{cls}^u(\theta; \mathcal{D}) + \mathcal{L}_r(\theta; \mathcal{D})$.

**Bias in Prototype Classifier.** Fig. 1a shows that the baseline exhibits pseudo-label bias (blue curve) in seen samples in unlabeled data. From Eq. (2), we see that the pseudo-label $\boldsymbol{p}_T(\boldsymbol{x}_i)$ is generated by calculating the similarity between a sample's class token and class prototypes. If the prototypes are too similar, this lack of discrimination may lead to incorrect labels. These errors in the teacher network's pseudo-labels can reinforce biases and cause further misclassifications in the student network through Eq. (3). Without intervention, the model remains stuck in these biases, as indicated by the persistent bias shown in the blue curve in Fig. 1a.

To address this bias, the common approach is to improve the discriminative power of class prototypes. However, we can't directly modify the classifier weights. Instead, we focus on adjusting the pseudo-labels $\boldsymbol{p}(\boldsymbol{x}_i)$ generated by the prototypes, allowing back-propagation to update the classifier and optimize the class prototypes.

## 4 DEBIASED IMBALANCED PSEUDO-LABELING

In this section, we propose an effective debiased method for GCD, DebiasGCD, as motivated in Sec. 1 and illustrated in Fig. 2. Concretely, we describe the proposed Dynamic Prototype Debiasing (DPD) in Sec. 4.1 and Local Representation Alignment (LRA) modules in Sec. 4.2, respectively.

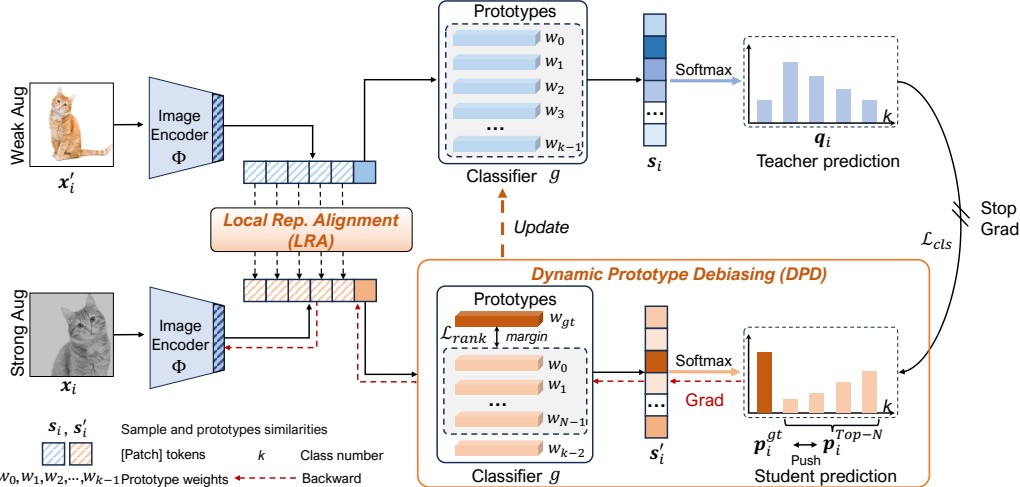

Figure 2: The overall framework of DebiasGCD. First, DPD enhances prototype discriminability using $\mathcal{L}$rank to push the ground-truth prototype $\boldsymbol{w}_{gt}$ and Top-$N$ smallest probability prototypes $[\boldsymbol{w}_0, \boldsymbol{w}_1, \ldots, \boldsymbol{w}_{k-1}]$ to maintain a margin. Next, LRA aligns patch tokens from Image Encoder, enriching class tokens sent to prototypes and facilitating DPD to discover more category-specific features. Finally, teacher network prototypes are updated by those from the student network.

## 4.1 DYNAMIC PROTOTYPE DEBIASING

**Motivation.** Although the prototype classifier learning (Wen et al., 2023) has alleviated the costly clustering process and improved accuracy compared to GCD (Vaze et al., 2022b), it encounters a bias in generating imbalanced pseudo-labels for semi-supervised learning, as shown in Fig. 1a and Fig. 1b. This bias arises because SimGCD relies solely on contrastive learning (see Sec. 3) and does not directly optimize prototypes for pseudo-label generation. Therefore, we argue that enhancing the discriminability of each prototype is essential. To tackle this, we propose the Dynamic Prototype Debiasing (DPD) strategy, which dynamically augments class-specific discriminability among prototypes to mitigate bias in a pseudo-label generation.

Inspired by Wang et al. (2022), who use adaptive margin loss to reduce bias in semi-supervised learning, we adopt a similar approach to improve the separation between prototypes. However, instead of generating class-balanced logits as in Wang et al. (2022), we take a more granular approach by applying prototype ranking loss. Specifically, we enforce this ranking-based loss on labeled data to dynamically increase the margin between prototypes, enhancing their ability to distinguish between classes. This enhanced discriminability in the classification of our method is shown in Fig. 3.

Formally, let $\boldsymbol{p}_S(\boldsymbol{x}_i) = \left[p_i^0, p_i^1, \ldots, p_i^{k-1}\right]$ from Eq. (2) represents the probability vector for sample $\boldsymbol{x}_i$ in $\mathcal{D}^l$ from *student* network, where $k$ is the number of categories. As shown in Fig. 2, We employ a margin ranking loss to rank the ground-truth prototype $\boldsymbol{w}_{gt}$ and the Top-$N$ smallest prototypes $\boldsymbol{w}_n$ based on the probabilities from the prototype classifier (see Sec. 3). First, we obtain the remaining probabilities except for the ground-truth ones $\boldsymbol{p}_S^{gt}(\boldsymbol{x}_i) = \left[p_i^{gt}\right]$, and sort these probabilities from smallest to largest, resulting in $\boldsymbol{p}_S^{reversed}(\boldsymbol{x}_i) = \left[p_i^0, p_i^1, \ldots, p_i^{(k-2)}\right]$. Subsequently, we select the Top-$N$ smallest probability within one sample, denoted as $\boldsymbol{p}_S^{Top-N}(\boldsymbol{x}_i)$, where $N$ is a hyperparameter and $N \leq k-1$. Then, we apply the margin ranking loss as follows:

$$\mathcal{L}_{rank}(\theta; \mathcal{D}^l) = \frac{1}{|\mathcal{D}^l|} \sum_{\boldsymbol{x}_i \in \mathcal{D}^l} \max\left(0, -r\left(\boldsymbol{p}_S^{gt}(\boldsymbol{x}_i) - \boldsymbol{p}_S^{Top-N}(\boldsymbol{x}_i)\right) + margin\right) \quad (4)$$

where $\mathcal{D}^l$ is labeled data, $r$ is a ranking label ( 1 or -1), setting $r=1$ assume $\boldsymbol{p}_S^{gt}(\boldsymbol{x}_i)$ rank higher than $\boldsymbol{p}_S^{Top-N}(\boldsymbol{x}_i)$. The *margin* is a pre-defined boundary value. If the distance between $\boldsymbol{w}_i^{gt}$ and $\boldsymbol{w}_i^{gt}$ is less than *margin*, the loss is 0; otherwise, the loss increases linearly. Minimizing $\mathcal{L}_{rank}(\theta; \mathcal{D}^l)$

ensures a margin between the correct prototype $\boldsymbol{w}_{gt}$ and the lowest-ranked $\boldsymbol{w}_N$ in Top-$N$ probabilities, improving prototype discriminability and pseudo-label accuracy.

## 4.2 LOCAL REPRESENTATION ALIGNMENT

It's worth noting that the previous prototype classifiers (Wen et al., 2023; Wang et al., 2024) relied solely on global representations in semi-supervised learning, neglecting local detail supervision, which is crucial in fine-grained datasets like CUB. For instance, birds with similar global features (e.g., color, shape) can be misclassified, reducing pseudo-label quality. In contrast, our approach encourages learning from local features, such as bill shape variations, to enhance prototype learning and improve classifier discriminability.

We propose a Local Representation Alignment (LRA) module to maintain semantic consistency of local features, improving the class token's ability to gather global information in the attention mechanism and enhancing the quality of pseudo-labels. Given an input $\boldsymbol{x}_i$, its patch tokens are all features except the first row vector of class token in its feature representation $\boldsymbol{z}_i$, denoted as $\boldsymbol{z}_i^{feat} \in \mathbb{R}^{n \times d}$, $n$ is patch tokens as explained in the architecture on Sec. (3), $d$ is feature dimension. Next, we apply softmax on $\boldsymbol{z}_i^{feat}$ to the feature dimension:

$$\boldsymbol{p}^{feat}(\boldsymbol{x}_i) = \frac{\exp\left(\boldsymbol{z}_i^{feat}/\tau\right)}{\sum_{j=0}^{d-1} \exp\left(\boldsymbol{z}_{i,j}^{feat}/\tau\right)}, \tag{5}$$

The feature distribution $\boldsymbol{p}^{feat}(\boldsymbol{x}_i')$ is produced from another sample $\boldsymbol{x}_i'$, using a sharper temperature $\tau$. Similar to the classification in Eq. (3), the LRA loss applies a simple cross-entropy loss between weak and strong views:

$$\mathcal{L}_{patch}(\theta; \mathcal{D}) = -\frac{1}{|\mathcal{D}|} \sum_{\boldsymbol{x}_i \in \mathcal{D}} \left\langle \boldsymbol{p}_T^{feat}\left(\boldsymbol{x}_i\right), \log\left(\boldsymbol{p}_S^{feat}\left(\boldsymbol{x}_i'\right)\right)\right\rangle + \left\langle \boldsymbol{p}_T^{feat}\left(\boldsymbol{x}_i'\right), \log\left(\boldsymbol{p}_S^{feat}\left(\boldsymbol{x}_i\right)\right)\right\rangle \tag{6}$$

As we know, the class token for classification aggregates information about patch tokens through the attention mechanism (Dosovitskiy et al., 2021): $C_i' = \sum_{j=0}^{N-1} \alpha_{ij} P_j$, where $\alpha_{ij} = \frac{\exp\left(\frac{C_i \cdot P_j^T}{\sqrt{d}}\right)}{\sum_{m=0}^{N-1} \exp\left(\frac{C_i \cdot P_m^T}{\sqrt{d}}\right)}$,

Where $C$ and $P$ are short for class token and patch token, respectively. The alignments make local features (patch tokens $P$) become more consistent in the feature space, making it easier to identify local features of the same object or scene in different views, thus enhancing global semantic (class tokens $C'$) extraction.

In summary, the LRA module aligns the local representations of two view samples, allowing class tokens to aggregate detailed feature information and thus enhancing the DPD approach for a more discriminative classifier.

**Overall Loss.** we combine the classification loss (Eq. (3)) and local representation alignment loss (Eq. (6)) in self-distillation. The ultimate supervision loss in prototype classifier training is updated as $\mathcal{L}_{self-dis} = \underbrace{\mathcal{L}_{cls}^s(\theta; \mathcal{D}^l) + \mathcal{L}_{cls}^u(\theta; \mathcal{D})}_{\text{Baseline}} + \alpha \cdot \mathcal{L}_{patch}(\theta; \mathcal{D})$, where $\mathcal{L}_{cls}^s(\theta; \mathcal{D}^l)$ is a standard cross-entropy loss on labeled data, and $\mathcal{L}_{cls}^u(\theta; \mathcal{D})$ is proposed in Eq. (3).

By simply integrating the Dynamic Prototype Debiasing (DPD) and Local Representation Alignment (LRA) modules, we propose a debiased pseudo-labeling in GCD (DebiasGCD). The overall loss for debiasing is formulated as:

$$\mathcal{L} = \underbrace{\mathcal{L}_{rep}(\theta; \mathcal{D}) - H(\theta; \mathcal{D})}_{\text{Baseline}} + \underbrace{\mathcal{L}_{self-dis}(\theta; \mathcal{D})}_{\text{Updated}} + \beta \cdot \mathcal{L}_{rank}(\theta; \mathcal{D}^l) \tag{7}$$

where $\alpha$, $\beta$ are balance factors controlling the prototypical classifier learning, and $H(\theta; \mathcal{D})$ represents the mean-entropy-maximisation regulariser (Assran et al., 2022). Notably, the balance factors for the baseline losses are the same as those of SimGCD. The algorithm in the appendix describes one training step of our proposed DebiasGCD.

Table 1: Datase overview for GCD, containing the specific classes ('Old' and 'New') and corresponding images of labeled ($\mathcal{D}^l$) and unlabeled sets ($\mathcal{D}^u$).

| Dataset | | CUB | SCars | FGCV-Aircraft | CIFAR-10 | CIFAR-100 | ImageNet-100 |
|---|---|---|---|---|---|---|---|
| Labeled $\mathcal{D}^l$ | Old | 100 | 98 | 50 | 5 | 50 | 50 |
| | Images | 1.5k | 2.0k | 1.7k | 12.5k | 20.0k | 31.9k |
| Unlabeled $\mathcal{D}^u$ | New | 200 | 196 | 100 | 10 | 100 | 100 |
| | Images | 4.5k | 6.1k | 5.0k | 37.5k | 30.0k | 95.3k |

# 5 EXPERIMENTS

## 5.1 EXPERIMENTAL SETUP

**Datasets.** We evaluate the effectiveness of our approach on six datasets, including three generic object recognition datasets (namely CIFAR-10/100 (Krizhevsky et al., 2009) and ImageNet-100 (Tian et al., 2020)) and three semantic shift benchmarks (SSB) (Vaze et al., 2022b) as well as fine-grained datasets: CUB (Wah et al., 2011), Stanford Cars (Krause et al., 2013), and FGVC-Aircraft (Maji et al., 2013). Following GCD (Vaze et al., 2022b), we randomly sub-sampling 50% of the seen categories within the training set to construct labeled set $\mathcal{D}^l$, with the remaining seen and novel category images constituting the unlabeled subset $\mathcal{D}^u$. Table 1 details our experimental split protocol.

**Evaluation protocols.** Following GCD (Vaze et al., 2022b) evaluation protocol, we employ clustering accuracy (ACC) to evaluate the model performance across all datasets. Concretely, ACC is computed using the predicted labels $\hat{y}$ with ground truth labels $y^*$, defined as $ACC = \frac{1}{M} \sum_{i=1}^{M} \mathbb{1}\left(y_i^* = p\left(\hat{y}_i\right)\right)$, where $M = |\mathcal{D}^u|$, and $p$ aligns predicted cluster assignments with ground truth class labels using the Hungarian optimal assignment algorithm (Kuhn, 1955).

**Implementation details.** Following GCD works (Vaze et al., 2022b; Wen et al., 2023), we use a ViT-B/16 backbone (Dosovitskiy et al., 2021), pre-trained with DINO (Caron et al., 2021). During training, we fine-tune only the last attention block of the backbone across all datasets. We adopt *strong/weak* data augmentation strategies (Sohn et al., 2020) as outlined in previous research (Vaze et al., 2023), applying strong augmentation to the student network and weak augmentation to the teacher network. Our setup includes a batch size of 128, 200 training epochs, and an initial learning rate of 0.1 with cosine decay. During classifier training, we initial $\tau = 0.07$, warming up to 0.04 within the first 30 epochs. Parameters $r$ and *margins* are set to 1 in Eq. (4), and we use a random seed of 1. Experiments are implemented in PyTorch on Nvidia Tesla V100 GPUs.

## 5.2 QUANTITATIVE COMPARISON

Table 2: Classification results on SSB (Vaze et al., 2022a) and generic object recognition datatsets. **Bold** represents the best results, underline is the second-best. $\Delta$ denotes margins ahead of SimGCD.

| Methods | CUB | | | Stanford Cars | | | FGVC-Aircraft | | | CIFAR-10 | | | CIFAR-100 | | | ImageNet-100 | | |
|---|---|---|---|---|---|---|---|---|---|---|---|---|---|---|---|---|---|---|
| | All | Old | New | All | Old | New | All | Old | New | All | Old | New | All | Old | New | All | Old | New |
| $k$-means MacQueen et al. (1967) | 34.3 | 38.9 | 32.1 | 12.8 | 10.6 | 13.8 | 16.0 | 14.4 | 16.8 | 83.6 | 85.7 | 82.5 | 52.0 | 52.2 | 50.8 | 72.7 | 75.5 | 71.3 |
| RankStats+ Han et al. (2022) | 33.3 | 51.6 | 24.2 | 28.3 | 61.8 | 12.1 | 26.9 | 36.4 | 22.2 | 46.8 | 19.2 | 60.5 | 58.2 | 77.6 | 19.3 | 37.1 | 61.6 | 24.8 |
| UNO+ Fini et al. (2021) | 35.1 | 49.0 | 28.1 | 35.5 | 70.5 | 18.6 | 40.3 | 56.4 | 32.2 | 68.6 | 98.3 | 53.8 | 69.5 | 80.6 | 47.2 | 70.3 | 95.0 | 57.9 |
| ORCA Liu et al. (2023) | 35.3 | 45.6 | 30.2 | 23.5 | 50.1 | 10.7 | 22.0 | 31.8 | 17.1 | 81.8 | 86.2 | 79.6 | 69.0 | 77.4 | 52.0 | 73.5 | 92.6 | 63.9 |
| GCD Vaze et al. (2022b) | 51.3 | 56.6 | 48.7 | 39.0 | 57.6 | 29.9 | 45.0 | 41.1 | 46.9 | 91.5 | **97.9** | 88.2 | 73.0 | 76.2 | 66.5 | 74.1 | 89.8 | 66.3 |
| GPC Zhao et al. (2023) | 52.0 | 55.5 | 47.5 | 38.2 | 58.9 | 27.4 | 43.3 | 40.7 | 44.8 | 90.6 | 97.6 | 87.0 | 75.4 | **84.6** | 60.1 | 75.3 | 93.4 | 66.7 |
| XCon Fei et al. (2022) | 52.1 | 54.3 | 51.0 | 40.5 | 58.8 | 31.7 | 47.7 | 44.4 | 49.4 | 96.0 | 97.3 | 95.4 | 74.2 | 81.2 | 60.3 | 77.6 | 93.5 | 69.7 |
| PromptCAL Zhang et al. (2023b) | 62.9 | 64.4 | 62.1 | 50.2 | 70.1 | 40.6 | 52.2 | 52.2 | 52.3 | 97.9 | 96.6 | 98.5 | 81.2 | 84.2 | 75.3 | 83.1 | 92.7 | 78.3 |
| $\mu$GCD Vaze et al. (2023) | 65.7 | 68.0 | 64.6 | 56.5 | 68.1 | 50.9 | 53.8 | 55.4 | 53.0 | - | - | - | - | - | - | - | - | - |
| DCCL Pu et al. (2023) | 63.5 | 60.8 | **64.9** | 43.1 | 55.7 | 36.2 | - | - | - | 96.3 | 96.5 | 96.9 | 75.3 | 76.8 | 70.2 | 80.5 | 90.5 | 76.2 |
| SimGCD Wen et al. (2023) | 60.3 | 65.6 | 57.7 | 53.8 | 71.9 | 45.0 | 54.2 | 59.1 | 51.8 | 97.1 | 95.1 | 98.1 | 80.1 | 81.2 | 77.8 | 83.0 | 93.1 | 77.9 |
| DebiasGCD (Ours) | **67.4** | **76.3** | 63.0 | **61.8** | **78.9** | **53.6** | **56.8** | **65.7** | **52.3** | 97.4 | 95.4 | 98.4 | **84.1** | 84.2 | **84.0** | **84.2** | **94.0** | **79.3** |
| $\Delta$ | +7.1 | +10.7 | +5.2 | +8.0 | +7.0 | +8.6 | +2.6 | +6.6 | +0.5 | +0.3 | +0.3 | +0.3 | +4.0 | +3.0 | +6.2 | +1.2 | +0.9 | +1.4 |

**Comparison with Baseline.** We compare our DebiasGCD with other methods on both SSB and generic object recognition datasets, as shown in Table 2. Identifying fine-grained samples in GCD is challenging due to their subtle differences, such as similar heads with different-sized bills. Excitingly, our method outperforms SimGCD on fine-grained datasets. Specifically, on CUB and Stanford Cars, it achieves a **10.7%/5.2%** and **7.0%/8.6%** improvement on 'Old' and 'New' classes, respectively. Additionally, DebiasGCD performs well on generic object recognition datasets, with improvements of

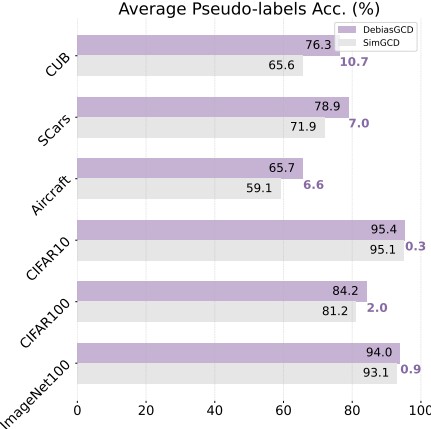

Figure 4: DebiasGCD achieves higher pseudo-label accuracy than SimGCD across all datasets.

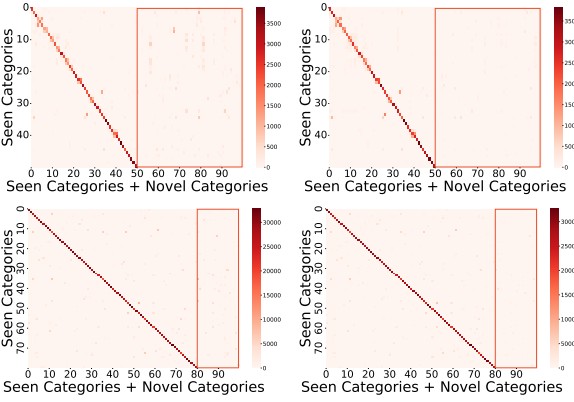

Figure 5: Pseudo-labeling in SimGCD (Column 1, with bias) and our method (Column 2, with less noise) for Aircraft and CIFAR-100. Note that there are 80 seen categories and 20 novel categories in CIFAR-100.

0.5% on CIFAR-10, 4.0% on CIFAR-100, and 1.2% on ImageNet-100 for 'All' classes. These results demonstrate that DebiasGCD successfully captures local representation features, generating high-quality pseudo-labels for prototypical learning. Furthermore, Fig. 4 shows that the pseudo-labeling accuracy increases significantly after debiasing across all datasets, and Fig. 5 indicates our debiased method recorrects pseudo-labeling effectively in FGVC-Aircraft and CIFAR-100, respectively.

**Visualizations of Feature and Attention Distributions.** Firstly, We first use t-SNE to visualize the feature space of five randomly selected categories from CIFAR-100, both seen and novel. As shown in Fig. 3, our debiasing approach creates clearer margins and tighter clusters compared to DINO and SimGCD, demonstrating better classification discriminability. Secondly, we visualize the attention maps of different attention heads in the image encoder $\Phi$ to show their focus on various image regions. As shown in Fig. 6, compared to the baseline SimGCD, our method highlights class-specific object parts while reducing background noise, indicating that DebiasGCD effectively enhances local representation learning.

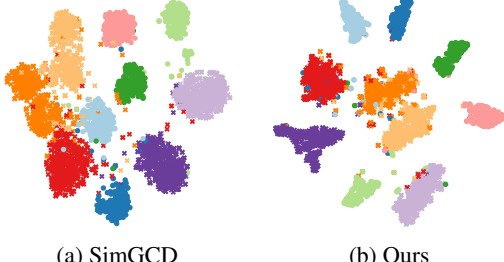

(a) SimGCD      (b) Ours

Figure 3: t-SNE visualization of 10 randomly sampled classes from the CIFAR-100. ● denotes seen classes, while ✖ represents novel classes. Our proposed DebiasGCD demonstrates fewer misclassified samples in both seen and novel categories compared to the baseline SimGCD.

### 5.3 ABLATION STUDY

To examine the contributions of various elements of our proposed approach, we conduct extensive experiments on both SSB and generic object recognition datasets, as shown in Tables 3, 4 and Fig. 7.

**Effect of Dynamic Prototype Debiasing (DPD).** Rows (1) in Table 3 illustrate the impact of incorporating DPD. This technique significantly improves accuracy, especially when combined with LRA. Comparing Row (1) with SimGCD (Row (0)), DPD enhances performance in 'New' classes for CIFAR-10 and CIFAR-100 and improves all metrics ('All', 'Old', and 'New') for CUB and Stanford Cars. These results highlight the efficacy of DPD in improving model performance.

**Effect of Local Representation Aligning (LRA).** Rows (2) also show the impact of introducing LRA. Like DPD, LRA enhances performance in 'New' classes for CIFAR-10 and CIFAR-100 and improves all metrics for CUB and Stanford Cars when comparing Row (2) with SimGCD (Row (0)). Additionally, LRA further improves performance across 'All' classes by 3.9%, 3.8%, 1.9%, and 6.9%

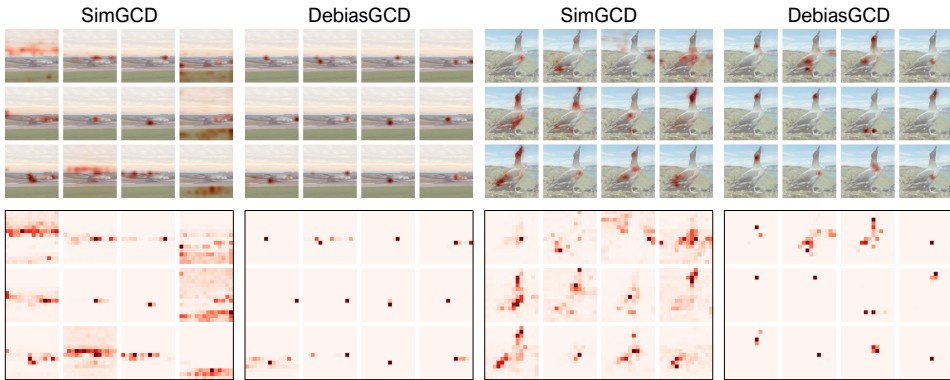

Figure 6: Attention visualization of 12 different attention heads in the final layer of the image encoder on FGVC-Aircraft and CUB, respectively.

Table 3: Ablation study on the different components of our approach. '$\mathcal{L}_{rank}$' and '$\mathcal{L}_{patch}$' are losses corresponding to the proposed DPD in Eq. (4) and LRA in Eq. (6), respectively. Row (0) shows the baseline SimGCD results.

| Index | Component | | CUB | | | SCars | | | CIFAR-100 | | |
|---|---|---|---|---|---|---|---|---|---|---|---|
| | $\mathcal{L}_{rank}$ | $\mathcal{L}_{patch}$ | All | Old | New | All | Old | New | All | Old | New |
| (0) | ✗ | ✗ | 60.3 | 65.6 | 57.7 | 53.8 | 71.9 | 45.0 | 80.1 | 81.2 | 77.8 |
| (2) | ✓ | ✗ | 62.0 | 66.8 | 59.6 | 57.8 | 77.4 | 47.6 | 82.0 | 82.7 | 81.3 |
| (3) | ✗ | ✓ | 63.5 | 68.5 | 61.0 | 58.0 | 77.0 | 48.9 | 81.1 | 82.6 | 78.0 |
| (5) | ✓ | ✓ | **67.4** | **76.3** | **63.0** | **61.8** | **78.9** | **53.6** | **84.2** | **84.2** | **84.0** |

in all datasets combined with DPD, as shown by comparing Rows (2) and (3). This also indicates LRA's effectiveness when paired with DPD.

**Effect of Top $N$ in Dynamic Prototype Debiasing.** In Sec. 4.1, we propose a debiased imbalanced pseudo-labeling strategy via a ranking-based prototype loss (Eq. 4). Fig. 7 shows the performance across different Top $N$ values on all datasets. Specifically, $N = 0$ indicates the performance of the baseline without debiasing. When $N = 1$, we select the smallest value in the probability vector $\boldsymbol{p}(\boldsymbol{x}_i)$, resulting in $\boldsymbol{p}^{reversed}(\boldsymbol{x}_i) = \left[p_i^0\right]$ for a sample $\boldsymbol{x}_i$ in Eq. (4). Compared to SimGCD ($N = 0$), our method improves consistently performance in all classes ('All', 'Old', and 'New') for various $N$ values in CUB, Stanford Cars, CIFAR-10, and CIFAR-100 datasets.

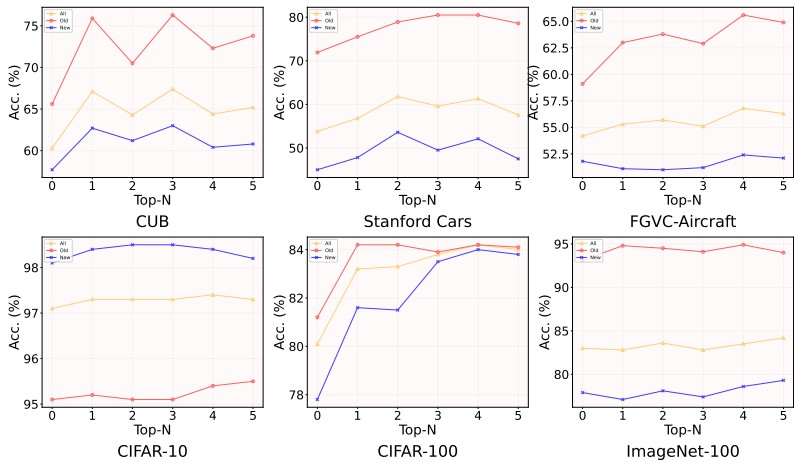

Figure 7: Ablation on the Top-$N$ in DPD in SSB and generic object recognition datasets.

**Effect of Hyperparameters $\alpha$ and $\beta$.** Table 4 outlines the values of $\alpha$ and $\beta$ used in Eq. (7). The parameter $\alpha$ regulates the influence of detailed feature learning on the prototype classifier, while $\beta$ controls the balance of prototype margins within the DPD framework. Here, we fix $\alpha = 1$ for local representation alignment and adjust $\beta$ to optimize margin balancing. As shown in the table, a setting of $\beta = 1$ delivers superior performance for CUB and Aircraft, whereas $\beta = 0.5$ yields better results for Stanford Cars, CIFAR-10, CIFAR-100, and ImageNet-100. Consequently, we set $\beta = 1$ for CUB and Aircraft, and $\beta = 0.5$ for the remaining datasets.

Table 4: Ablation study on $\alpha$ and $\beta$ of Eq. (7) in SSB and generic object recognition datasets.

| $\alpha$ | $\beta$ | CUB | | | Stanford Cars | | | FGCV-Aircraft | | | CIFAR-10 | | | CIFAR-100 | | | ImageNet-100 | | |
|---|---|---|---|---|---|---|---|---|---|---|---|---|---|---|---|---|---|---|---|
| | | All | Old | New | All | Old | New | All | Old | New | All | Old | New | All | Old | New | All | Old | New |
| 1 | 0.5 | 61.1 | 70.0 | 56.7 | **61.8** | 78.9 | **53.6** | 54.9 | 67.4 | 48.7 | **97.4** | 95.4 | **98.4** | **84.2** | 82.5 | **83.6** | **80.4** | 94.0 | **79.3** |
| | 1.0 | **67.4** | **76.3** | **63.0** | 58.9 | **81.7** | 47.9 | **56.8** | **65.7** | **52.4** | 97.3 | **96.7** | 97.6 | 81.9 | **84.9** | 75.9 | 83.3 | **94.5** | 77.6 |

## 5.4 EXTEND TO OTHER OPEN-WORLD WORK

It is worth noting that our proposed method serves as a plug-and-play solution that can be seamlessly integrated with other open-world works, such as LegoGCD (Cao et al., 2024) and SPTNet (Wang et al., 2024). Specifically, we incorporate the proposed DPD and LRA modules into the above frameworks in fine-grained datasets CUB, Stanford Cars, and generic dataset CIFAR-100. Table 5 shows that our components enhance classification performance in LegoGCD across all datasets, particularly improving the accuracy on 'New' categories by 3.2% and 5.1% in the fine-grained CUB and Stanford Cars datasets, respectively. Furthermore, our modules contribute to an increase in overall ('All') accuracy in SPTNet, with gains of 1.0% and 3.0% on the CUB and Stanford Cars datasets. Although the performance on the CIFAR-100 dataset has slightly decreased (-0.1 in 'All'), we believe it can be improved with further parameter adjustments. Overall, our proposed method is effective and can be applied to other open-world scenarios.

Table 5: Classification results on other open-world works combined with our proposed components across fine-grained and generic recognition datasets. $\Delta$ indicates margins after debiasing.

| Index | Methods | CUB | | | Stanford Cars | | | CIFAR-100 | | |
|---|---|---|---|---|---|---|---|---|---|---|
| | | All | Old | New | All | Old | New | All | Old | New |
| (1) | LegoGCD Cao et al. (2024) | 63.8 | 71.9 | 59.8 | 57.3 | 75.7 | 48.3 | 81.8 | 81.4 | 82.5 |
| (2) | With debiasing | 66.7 | 72.1 | 64.0 | 60.9 | 76.1 | 53.4 | 82.6 | 84.2 | 81.1 |
| (3) | $\Delta$ | **2.9** | **+0.2** | **+3.2** | **+3.6** | **+0.4** | **+5.1** | **+0.8** | **+2.8** | -1.4 |
| (4) | SPTNet Wang et al. (2024) | 65.8 | 68.8 | 65.1 | 59.0 | 79.2 | 49.3 | 81.3 | 84.3 | 75.6 |
| (5) | With debiasing | 66.8 | 73.1 | 63.7 | 62.0 | 78.0 | 54.4 | 81.2 | 84.0 | 76.0 |
| (6) | $\Delta$ | **1.0** | **+4.3** | -1.4 | **+3.0** | -1.2 | **+4.1** | -0.1 | -0.3 | **+0.4** |

## 6 CONCLUSION

In this paper, we first investigate the previously unrecognized bias of imbalanced pseudo-labeling in the GCD task. We then propose an effective debiasing method, DebiasGCD, to address this imbalance between seen and novel categories in classifier prototype learning. To implement this debiasing, we propose a dynamic prototype debiasing technique to maintain a margin between prototypes dynamically, encouraging the network to explore category-specific features and enhance prototype distinction. Furthermore, to improve the learning for more discriminable representations in DPD, we propose a local representations alignment module to discover more subtle features that benefit classification, especially in fine-grained datasets. Extensive results show that our DebiasGCD significantly outperforms the baseline SimGCD, effectively mitigating the pseudo-labeling bias.

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
