# Supplementary Material for Debiased Imbalanced Pseudo-Labeling for Generalized Category Discovery

## Contents

# 1 PSEUDO CODES

The following algorithm outlines the training process of our DebiasGCD method. Each variable and formula are annotated, and every function is linked to the specific equations presented in our paper.

---
**Algorithm 1** Pseudo code on one step for DebiasGCD
---

```
1  #x1, x2: two view samples
2  #s_proj, s_cls, t_cls, s_patch, t_patch: projection feature, logits (
       similarities), and patch tokens for student and teacher
3  #label, mask: image label and corresponding mask
4
5  def training_step(x1, x2):
6      s_proj, s_cls, s_patch = model([x1, x2])
7      t_cls = s_cls.detach()
8      t_patch = s_patch.detach()
9
10     #Representation learning
11     loss_{rep} = contrastive_learning(s_proj, label, mask)
12
13     #Regularization loss
14     loss_reg = mean_max_entropy(s_cls)
15
16     #Classification uses ground-truth labels on labeled data
17     loss^l_{cls} = cross_entropy(t_cls, s_cls, label=target[mask=1])
18
19     #Classification using pseudo-labels in all data
20     loss^{u}_{cls} = entropy(t_cls, s_cls) #Eq.(3)
21
22     #DPD loss in #Eq.(4)
23     topn_val = Top-N(s_cls[label=target[mask=1], largest=False)
24     loss_{rank} = margin_rank(s_cls[label=target[mask=1]], topn_val)
25
26     #LRA loss
27     loss_{patch} = entropy(t_patch, s_patch) #Eq.(6)
28
29     #Self-distillation
30     loss_{self-dis} = loss^l_{cls} + loss^{u}_{cls} + loss_{patch}
31
32     # Overall loss
33     loss = loss_{rep} + loss_{self-dis} + loss_{rank} - loss_reg # Eq.(7)
34
35     return loss
```

---

# 2 DESCRIPTION OF REPRESENTATION LEARNING

This is the whole representation learning in GCD work. Formally, given two views $x_i$ and $x_i'$ of the same image in a all data $\mathcal{D}$, the parameters of feature extractor $\Phi$ can be updated by the InfoNCE loss (van den Oord et al., 2018) in self-supervised contrastive learning:

$$\mathcal{L}_{rep}(\theta; \mathcal{D}) = -\frac{1}{|\mathcal{D}|} \sum_{\boldsymbol{x}_i \in \mathcal{D}} \log \frac{\exp \left\langle \boldsymbol{z}_i^{cls}, \boldsymbol{z}_i^{cls\prime} \right\rangle / \tau}{\sum_n^{n \neq i} \exp \left\langle \boldsymbol{z}_i^{cls}, \boldsymbol{z}_n^{cls} \right\rangle / \tau} \tag{1}$$

where $\boldsymbol{z}^{cls}$ is the first row vector in its feature representation $\boldsymbol{z}_i = h \circ (\Phi(x_i))$, denoted as $\boldsymbol{z}_i^{cls}$. $\tau$ is a temperature hyperparameter. Analogous to Eq. (1), we use supervised contrastive loss in the same class as:

$$\mathcal{L}_{rep}(\theta; \mathcal{D}^l) = -\frac{1}{|\mathcal{D}^l|} \sum_{\boldsymbol{x}_i \in \mathcal{D}} -\frac{1}{|\mathcal{N}(i)|} \sum_{q \in \mathcal{N}(i)} \log \frac{\exp \left\langle \boldsymbol{z}_i^{cls}, \boldsymbol{z}_q^{cls} \right\rangle / \tau}{\sum_n^{n \neq i} \exp \left\langle \boldsymbol{z}_i^{cls}, \boldsymbol{z}_n^{cls} \right\rangle / \tau} \tag{2}$$

Finally, the total contrastive loss on the model's representation is given as:

$$\mathcal{L}_{rep}(\theta; \mathcal{D}) = (1 - \lambda_1)\,\mathcal{L}_{rep}(\theta; \mathcal{D}^u) + \lambda_1 \mathcal{L}_{\text{rep}}(\theta; \mathcal{D}^l) \tag{3}$$

## 3 TRAINING TIME AND COMPUTATION COST

In this part, we compare the efficiency of time and resources used in current competitive works, such as LegoGCD and SPTNet. Table 1 shows the training time and computation costs in CUB datasets. Specifically, although SPTNet shows competitive performance, but it cost near and over 5 times in GPU memory and training time compared with our method, and our approach and baseline SimGCD, and LegoGCD all took about the same amount of time and computing resources. Therefore, our proposed method is efficient in classification, time, and computation cost.

Table 1: Comparison of computation costs among existing methods on CUB dataset. Our method is more efficient than SPTNet (Wang et al., 2024).

| Method | Epoch | Training Time | GPU usage (MiB) | All | Old | New |
|---|---|---|---|---|---|---|
| SimGCD Wen et al. (2023) | 200 | 4 hours and 35 mins | 5854 | 60.3 | 65.6 | 57.7 |
| LegoGCD Cao et al. (2024) | 200 | 4 hours and 40 mins | 5884 | 63.8 | 71.9 | 59.8 |
| SPTNet Wang et al. (2024) | 1000 | 23 hours and 21 mins | 29682 | 65.8 | 68.8 | 65.1 |
| Ours | 200 | 4 hours and 38mins | 6224 | 67.4 | 76.3 | 63.0 |