# OpenReview forum: "Debiased Imbalanced Pseudo-Labeling for Generalized Category Discovery"
_ICLR.cc/2025/Conference — ICLR 2025 Conference Withdrawn Submission_

### Official Review · Reviewer_W1Q5 · 2024-10-22

**Soundness:** 2
**Presentation:** 2
**Contribution:** 2
**Rating:** 3
**Confidence:** 5

**Summary:**

This paper tackles the problem of Generalized Category Discovery (GCD). The authors identify the easily overlooked bias in prototypical classifiers, namely the bias to misclassify known classes to the novel ones. To address this issue, the paper proposes DebiasGCD. Within it, dynamic prototype debiasing (DPD) explicitly maintains margins between classes. Besides, local representation alignment (LRA) is further proposed to ensure local alignment. Experiments in various settings show the method achieves remarkable performance.

**Strengths:**

1. This paper is well-motivated and easy to follow.
2. This paper delves into the easily overlooked bias in pseudo-labeling and addresses it through the proposed DPD and LRA.
3. The proposed method significantly improves the performance and brings about more balanced pseudo-labels.
4. This paper conducts comprehensive quantitative and qualitative visualizations to show the effectiveness of the method.

**Weaknesses:**

Major:
1. The novelty is relatively limited. The main learning components follow SimGCD, and ranking loss in DPD is not new. Plus, the proposed LRA is not fully explained (See question 2).
2. In LRA, Eq. (5), how can $z_i^{feat}$ be put into the $\exp$ operator, $z_i$ is a vector rather than a variable.
3. The bias *some seen categories are mistakenly classified as novel ones* is not very convincing to me. GCD is an imbalanced problem where some old classes samples are labeled, but new classes are purely unlabeled, in this way, models tend to bias some novel classes to old ones, which is revealed in SimGCD. However, in this paper, the bias is a little bit counterintuitive in the context of GCD.
4. Ablations on the margin value should be included.
5. Why r in Eq. (4) is 1 rather than -1. Plus, in Lines 268-269, there might be a typo in “if the distance between … and …”. Please clarify this point.

Minor:
1. Line 604. AGCD is a CVPR 2024 paper, please refine this reference.

**Questions:**

1. In DPD, why implement margin ranking loss with Top-N smallest prototypes, rather than the largest ones? Explanations should be provided.
2. Provide more details for Eq. (5), including the dimensions of related variables.

---

### Official Review · Reviewer_gHx6 · 2024-10-28

**Soundness:** 2
**Presentation:** 2
**Contribution:** 2
**Rating:** 3
**Confidence:** 5

**Summary:**

This paper focuses on the generalized category discovery problem. They first illustrate their motivation, debiasing the imbalanced pseudo-labeling, and propose a dynamic prototype debiasing module to debias the pseudo-label.  And they introduce LRA to align representation between two views. They conduct experiments to show the benefit of their method.

**Strengths:**

1. The motivation is clear.
2. Experiments with different open-world semi-supervised learning is convincing.

**Weaknesses:**

1. The notation in section 4.1 is quite confusing and difficult to understand. For instance, $ p_S^{gt} = [p_i^{gt}] $: does this represent the probability of the ground truth, or is it excluding the ground truth? Additionally, the process of ranking is unclear. It appears that the ranking loss is intended to enforce a uniform distribution of the prototypes, but this is not clearly explained.

2. The design of the patch loss is peculiar. For different augmentations, $ x_i $ and $ x_i^` $ may not be aligned within each patch. Furthermore, the design lacks thorough ablation studies. For example, what would the results be if a simple Mean Squared Error (MSE) loss were used instead?

3. In Table 2, comparisons with state-of-the-art methods [1,2] are missing. Additionally, experiments involving DINOv2 are not included.

4. As the paper focuses on imbalanced pseudo labels, more discussion about the imbalanced novel class discovery is needed.

[1] Rastegar et al. Learn to Categorize or Categorize to Learn? Self-Coding for Generalized Category Discovery. NeurIPS 2023.

[2] Gu et al. Class-relation Knowledge Distillation for Novel Class Discovery. ICCV2023.

**Questions:**

Please make the method more clear and easy to understand. And ablate the design of LRA. More importantly, please compare with [1,2] or state why there is no comparison with them.

---

### Official Review · Reviewer_sQbB · 2024-11-01

**Soundness:** 3
**Presentation:** 3
**Contribution:** 3
**Rating:** 6
**Confidence:** 3

**Summary:**

The paper addresses the Generalized Category Discovery (GCD) problem, highlighting a bias issue present in current prototype classifiers. To tackle this challenge, the paper propose a novel debiasing method called DebiasGCD. This framework comprises two key components: dynamic prototype debiasing (DPD), which expands the distances between prototypes, and local representation alignment (LRA), aimed at extracting more locally discriminative representation features.The proposed approach achieves a significant performance improvement over existing methods.

**Strengths:**

1. The paper introduces prototype ranking loss and local representation alignmnet to address the classifier bias problem in Generalized Category Discovery.
2. The paper is well-structured and well-presented, providing clear explanations for the motivations behind the approach.
3. The proposed framework demonstrates consistent performance gain over baseline methods on different benchmarks, and can serve as a plug-and-play module to intergrate with other existing methods.

**Weaknesses:**

1.Some analysis in experiments section lack explanation and details. For example, 1) The method is very sensitive to $\beta$ as showed in Table 4,will be good to discuss on this matter. 2) In prototype ranking loss, how to choose $r$ and margins need further clarification.

2.Herbarium is also a commonly used fine-grained dataset in GCD to evaluate model performance. However, this is missing in the experiments.

**Questions:**

1.As stated in weaknesses, the method is sensitive to some hyperparameters, especially on fine-grained benchmarks. I am looking forward to the authors' elucidation on this matter.

2.Additionaly, LegoGCD itself is a intergration module aimed for addressing old/novel class bias problem. I wonder if adding DebiasGCD can further help with correcting bias and improve pseudo label quality. it will be good to visualize the prediction bias comparison with LegoGCD (similiar to Fig 1).

---

### Official Review · Reviewer_YhAo · 2024-11-02

**Soundness:** 2
**Presentation:** 2
**Contribution:** 2
**Rating:** 5
**Confidence:** 5

**Summary:**

This submission introduces DebiasGCD, a method to improve Generalized Category Discovery (GCD) by addressing a classifier bias that mislabels seen categories as novel, leading to imbalanced pseudo-labeling. The approach integrates Dynamic Prototype Debiasing (DPD) and Local Representation Alignment (LRA) to enhance class prototype discrimination and local feature alignment, respectively. Extensive experiments across multiple datasets indicate DebiasGCD significantly enhances classification accuracy and pseudo-labeling reliability, particularly in fine-grained contexts.

**Strengths:**

1. The approach is well-supported by quantitative results showing improvements in pseudo-label accuracy and classification performance over strong baselines like SimGCD.
2.  The paper is generally well-written and easy to follow.
3. The GCD task is an important problem in open-world computer vision, and the authors' work has the potential to significantly improve the performance of GCD models.

**Weaknesses:**

1. The variation in results from the GCD benchmarks can be very large, so, as SimGCD does in its Supplementary Information, it is important to report all results along with error bars generated by 3 to 5 independent runs.
2. The performance of the method with DINOv2 should be explored, as it has been employed in recent GCD methods [A].
3. Further investigation is needed on the effectiveness of strong data enhancement.
4. As can be seen in Table 4, hyperparameter settings may require extensive optimization across different datasets, thus limiting scalability without further optimization guidance. This makes me more concerned about the real-world application of the method.
5. The paper contains several typos and formatting issues. (Errors in characters in the penultimate line of page 7, inconsistent font sizes in tables, and incorrect indexing of table 3)

[A] No Representation Rules Them All in Category Discovery. NeurIPS 2023.

**Questions:**

1. Could the authors provide more details on the hyperparameter tuning process for the DPD and LRA components, and how they determined the optimal values for the balance factors α and β? The paper mentions parameter adjustments as a solution to lower performance in some scenarios, but further guidelines would improve practical usability.

---

### Official Review · Reviewer_MY59 · 2024-11-03

**Soundness:** 2
**Presentation:** 2
**Contribution:** 2
**Rating:** 3
**Confidence:** 5

**Summary:**

This paper addresses the Generalized Category Discovery (GCD) task, which aims to categorize unlabeled data into known and novel classes with the knowledge learned from labeled data of known classes.
The authors propose a novel strategy named DebiasGCD, which incorporates a dynamic prototype debiasing (DPD) module to alleviate the bias in the pseudo label of labeled samples, and a local representation alignment (LRA) constraint to maintain the local consistency between contrastive augmented pair of samples.
The authors propose a novel DebiasGCD method that rectifies the pseudo label in the training process of GCD and shows significant improvements across six GCD benchmarks

**Strengths:**

S1. The proposed DPD in the DebiasGCD framework directly optimizes prototypes of known classes, which relieves the bias in pseudo-label that samples of known classes are mistakenly assigned to novel classes.
S2. The experiment section supports the effectiveness of the proposed DPD and LRA.

**Weaknesses:**

W1. Please check the classification ACC in Tab 4 carefully, there are obvious errors in the data, For example, β=0.5,All=80.4 Old=94.0 New=79.3, β=1.0,All=83.3 Old=94.5 New=77.6 for the ImageNet-100 dataset.

W2. Authors claim that DebiasGCD enables the model to generate cleaner pseudo labels for samples of known classes, while the interpretability of the proposed DPD is not sufficiently discussed and analyzed in the paper. For example, why does DPD choose the smallest prototype instead of the biggest one? Since the smallest prototype is relatively easy to distinguish from the GT classes by the model, which means these prototypes are hard to cause the bias in the pseudo label of known classes. In another paper that also tries to explicitly adjust the distance between prototypes, the authors choose the k-NN prototype as the negative data point [1]. Could the authors explain the motivation for this selection operation?

W3. Meanwhile, the authors seem to improperly mix the pseudo-label Acc and the classification Acc, as shown in Fig. 4, where the title is Pseudo-labels ACC, however, all values are the same as the classification ACC that is shown in Tab. 2. This accident raises concerns about Fig. 1(a)

W4. In Fig. 2, a strong augmentation is applied in the proposed DebiasGCD, but the effect of this operation is not shown in the ablation study.

W5. The setting of N for each dataset is not elaborated in the paper, and could the authors show the tendency of the DPD loss with the change of N?

W6. When N > 1, the dimensions of p_S^gt (x_i ) and p_S^(Top_N) (x_i) are not the same, how is the p_S^gt (x_i )-p_S^(Top_N) (x_i) in Eq. 4 realized?

W7. More works of GCD should be included in Tab 2 to demonstrate the effectiveness of the proposed methods, like CMS [2] and SPTNet [3]

W8. There are several typos in this paper, such as “p_( x_i)” in Line 209, and “w_i^gt,w_i^gt” in Line 269

[1] Fewer is more: A deep graph metric learning perspective using fewer proxies[J]. Advances in Neural Information Processing Systems, 2020
[2] Contrastive Mean-Shift Learning for Generalized Category Discovery[C]//Proceedings of the IEEE/CVF Conference on Computer Vision and Pattern Recognition. 2024.
[3] An Efficient Alternative Framework for Generalized Category Discovery with Spatial Prompt Tuning[C]//The Twelfth International Conference on Learning Representations.

**Questions:**

Please justify the issues in Weakness.

---

### Author Response · Authors · 2024-11-25
**Request for Manuscript Withdrawal with Gratitude**

Dear Reviewers,

Thank you for your time and valuable suggestions on our paper. After careful consideration, we have realized that our manuscript requires further revisions and improvements. We believe it would be best to withdraw the current submission and work on preparing a more comprehensive and complete version for resubmission in the future.

We are sincerely grateful for the time and effort you have invested in reviewing our work.

Thank you for your understanding.

Best regards,
Authors from 3711

---

### Note · Authors · 2024-11-25

I have read and agree with the venue's withdrawal policy on behalf of myself and my co-authors.